# Rational Construction of C@Sn/NSGr Composites as Enhanced Performance Anodes for Lithium Ion Batteries

**DOI:** 10.3390/nano13020271

**Published:** 2023-01-09

**Authors:** Guanhua Yang, Yihong Li, Xu Wang, Zhiguo Zhang, Jiayu Huang, Jie Zhang, Xinghua Liang, Jian Su, Linhui Ouyang, Jianling Huang

**Affiliations:** 1Guangxi Key Laboratory of Automobile Components and Vehicle Technology, Guangxi University of Science and Technology, Liuzhou 545006, China; 2Guangxi Key Laboratory of Low Carbon Energy Materials, Guangxi New Energy Ship Battery Engineering Technology Research Center, Guangxi Scientific and Technological Achievements Transformation Pilot Research Base of Electrochemical Energy Materials and Devices, School of Chemistry and Pharmaceutical Sciences, Guangxi Normal University, Guilin 541004, China

**Keywords:** Sn-based material, graphene, heteroatomic doping, carbon coating, lithium ion battery

## Abstract

As a potential anode material for lithium-ion batteries (LIBs), metal tin shows a high specific capacity. However, its inherent “volume effect” may easily turn tin-based electrode materials into powder and make them fall off in the cycle process, eventually leading to the reduction of the specific capacity, rate and cycle performance of the batteries. Considering the “volume effect” of tin, this study proposes to construct a carbon coating and three-dimensional graphene network to obtain a “double confinement” of metal tin, so as to improve the cycle and rate performance of the composite. This excellent construction can stabilize the tin and prevent its agglomeration during heat treatment and its pulverization during cycling, improving the electrochemical properties of tin-based composites. When the optimized composite material of C@Sn/NSGr-7.5 was used as an anode material in LIB, it maintained a specific capacity of about 667 mAh g^−1^ after 150 cycles at the current density of 0.1 A g^−1^ and exhibited a good cycle performance. It also displayed a good rate performance with a capability of 663 mAh g^−1^, 516 mAh g^−1^, 389 mAh g^−1^, 290 mAh g^−1^, 209 mAh g^−1^ and 141 mAh g^−1^ at 0.1 A g^−1^, 0.2 A g^−1^, 0.5 A g^−1^, 1 A g^−1^, 2 A g^−1^ and 5 A g^−1^, respectively. Furthermore, it delivered certain capacitance characteristics, which could improve the specific capacity of the battery. The above results showed that this is an effective method to obtain high-performance tin-based anode materials, which is of great significance for the development of new anode materials for LIBs.

## 1. Introduction

For the past few years, LIBs have played an important role in electric vehicles and portable electronic devices due to their high energy density [1,2,3,4], environmental friendliness and lack of the memory effect [5,6,7]. In addition, the demands for high-performance battery anode materials are increasing with the continuous development of technology. Nowadays, graphite is one of the commercial anode materials for LIBs [8,9,10], with the characteristics of low cost, good electronic conductivity, long cycle life and stable capacity. However, the capacity of commercial graphite is very close to the theoretical specific capacity of graphite (372 mAh g^−1^), and it is very difficult to improve the specific capacity of this kind of materials. In order to meet the development needs of high-performance LIBs, researchers need to carry out more in-depth research to explore and develop new high-performance anode materials [11,12].

Among the many potential new anode materials for LIBs, tin-based materials can form the alloy compound Li_4.4_Sn with high specific capacity. In particular, the theoretical specific capacity of metal tin is up to about 994 mAh g^−1^, which is more than two times higher than that of commercial graphite carbon materials [13]. In addition, tin has attracted researchers’ attention because of its abundant reserves, high safety and environmental friendliness [14,15], making it considered as a good material to replace the commercial graphite carbon anode of LIBs. However, tin is prone to cause huge volume expansion and contraction during the process of insertion and extraction of lithium, resulting in tin pulverization and its falling off from the current collector. This eventually leads to a serious reduction of the specific capacity and the deterioration of the cycle and rate performance of the electrode material [16,17,18]. In order to alleviate these problems, many researchers have conducted research on the nanolization of metal tin [15,18,19,20], the introduction of an active or inactive buffer matrix [21,22,23], etc. As a buffer matrix, carbon materials can effectively relieve the “volume effect” of metal tin and improve the specific capacity and cycle performance of the electrode material. Carbon materials have good mechanical properties, flexibility and high specific surface area, which not only is conducive to the preparation of composite materials, but also can stabilize the structure of metal tin and improve the cycle life of composite materials [24,25].

Graphene is a typical two-dimensional carbon nanomaterial with large specific surface area, good mechanical elasticity and superior electrical conductivity. Graphene can not only provide enough space to alleviate the volume expansion and contraction effect of metallic tin, but also improve the cyclic stability of the composite and reduce its internal resistance. Qin et al. [26] prepared Sn and graphene composite materials with NaCl as the template. The three-dimensional porous graphene network in the composite material can maintain the structure and interface stability of Sn nanoparticles, inhibit the aggregation of Sn nanoparticles and buffer the volume expansion, thus significantly enhancing the electrical conductivity and structural integrity of the overall electrode. Li et al. [27] prepared a Sn-graphene composite using a novel method of encapsulating Sn nanoparticles in graphene nanostructures by microwave plasma irradiation of SnO_2_. The results showed that the nanostructures of Sn-graphene were fully capable to prevent the volume changes and agglomeration effects of the Sn nanoparticles, successfully increasing the charging and discharging rates of the composite material. In addition, previous studies reported that the introduction of N and S created more active sites and increased the electrical conductivity of graphene, enabling the composite to exhibit high capacity and stability during cycling [28,29,30,31]. Jarulertwathana et al. [28] used a simple and low-cost method to synthesize high-performance tin/nitrogen-doped graphene-based nanocomposites, and the test results showed that the presence of pyridine nitrogen on the surface of nitrogen-doped graphene could improve the electrical conductivity of the composites. Although Sn/graphene can be prepared by different means, there are still great challenges to avoid pollution and achieve a low-cost and large-scale preparation. Therefore, it is important to explore and develop synthetic methods that are simple, fast and controllable to prepare Sn/graphene.

Herein, in order to easily prepare high-performance Sn/graphene composites, we propose to prepare a sulfur and nitrogen co-doped graphene solution by simple electrochemical exfoliation, then to anchor metal tin nanoparticles on the surface of the graphene by electroless plating and finally to use carbon coating to construct C@Sn/NSGr composites. The composite showed excellent structural characteristics, as the carbon coating limits the nanoparticles to a relatively small space, and the high conductivity and flexible graphene forms a network structure to further limit the tin nanoparticles, thus realizing a “double confinement” of metal tin. In addition, doping with heteroatoms such as sulfur and nitrogen can regulate the electronic properties of the material, accelerate electron transport and provide more active sites. When used in LIBs, the composites showed good electrochemical performance, thus providing a new strategy for the preparation of anode composites for LIBs with high performance.

## 2. Experimental

### 2.1. Preparation of Graphene

Graphene was prepared by electrochemical exfoliation, with a platinum sheet (20 mm × 20 mm) as the counter electrode and a graphite foil (30 mm × 60 mm) as the working electrode. They always maintained a distance of 1 cm during electrochemical exfoliation. Typically, the electrolyte (1 L) contained ammonium sulfate (0.1 mol L^−1^) and glycine (0.1 mol L^−1^), and the constant voltage was set at 12 V. After the complete exfoliation, the products were filtered and washed repeatedly with deionized water. Then, the collected products were dissolved in dimethyl formamide (250 mL) and sonicated for 2 h. In order to remove dimethyl formamide, the black suspension was centrifuged for 20 min at 3000 rpm. After that, the black solid was dispersed in deionized water, repeatedly washed and centrifuged for 20 min at 9000 rpm several times to obtain graphene. To obtain the surface-treated graphene, 0.5 g of graphene was dissolved in 100 mL of PdCl_2_ solution (0.1 g L^−1^) and sonicated for 20 min. After the ultrasonic treatment, the mixed solution was stirred for 30 min at room temperature. Then, 0.5 g of sodium phosphite was added into the above solution, which was stirred for 30 min. After filtration and washing, the surface-treated graphene was obtained.

### 2.2. Preparation of the C@Sn/NSGr Composites

The C@Sn/NSGr composites were prepared through electrochemical exfoliation, electroless tin plating and carbon coating treatment, as shown in Figure 1. Firstly, 20 g of thiourea and 2 g of sodium phosphite were dissolved in deionized water at 80 °C, and 1.5 g of SnCl_2_·2H_2_O (7.5 g L^−1^) was dissolved in hydrochloric acid (5 mL). Then, the SnCl_2_ solution and the sodium phosphite solution were added to the thiourea solution. After thoroughly mixing, the treated graphene and an appropriate amount of TritonX-100 were added. After that, deionized water was added into the solution to 200 mL, and the solution was stirred at 80 °C for 1 h. Then, the resulting black solid was washed and filtered with deionized water at 80 °C, and the Sn/NSGr-7.5 precursor was obtained after drying. In order to obtain the carbon-coated Sn/NSGr composite, glucose (70 mg) was dissolved in deionized water, and then the Sn/NSGr-7.5 precursor was added to the solution. The solution was stirred by a magnetic stirrer at 80 °C until the deionized water was completely evaporated. Finally, the obtained black product was annealed at 500 °C for 2 h in an argon atmosphere. The as-prepared product was called C@Sn/NSGr-7.5. Similarly, C@Sn/NSGr-5 and C@Sn/NSGr-10 were prepared with an identical procedure, using 5 g L^−1^ and 10 g L^−1^ of SnCl_2_·2H_2_O, respectively.

### 2.3. Characterization of the Materials

The crystallographic structures of the as-prepared samples were analyzed by X-ray diffraction (XRD) on Rigaku Ultima Ⅳ (Tokyo, Japan) with Cu Ka radiation at 40 kV. Scanning electron microscopy (SEM) on a Tescan MIRA LMS (Brno, Czech Republic) and transmission electron microscopy (TEM) on a JEOL JEM 2100F (Tokyo, Japan) were implemented to characterize the morphology of the materials. The distribution of the elements of the samples was investigated by an energy-dispersive spectroscope (EDS) attached to the SEM. In order to explore the chemical composition of the samples, X-ray photoelectron spectroscopy (XPS) tests were implemented on a Thermo Scientific K-Alpha system (New York, NY, USA). Nitrogen adsorption and desorption isotherms tests were carried on an ASAP 2020 Plus (Norcross, USA) by the Brunauer-Emmett-Teller (BET) method to investigate the specific surface area and pore structure of the materials.

### 2.4. Electrochemical Tests

All the as-prepared composite materials were used as active substances to produce electrodes. The preparation process of the electrodes was as follows. N-methyl pyrrolidone, Super-P and polyvinylidene fluoride were used as solvent, conductive additive and bonding agent, respectively. The slurry was composed of 80% active material, 10% Super-P and 10% polyvinylidene fluoride and was evenly coated on a copper foil. The pole sheets were placed in a vacuum oven for 6 h to dry and then cut into circular pieces with a diameter of 12 mm. The circular electrodes as the working electrodes and Li metal as the counter electrode were assembled into CR2025 cells in an argon-filled glovebox. The electrolyte contained 1M LiPF_6_, which was dissolved in a solution of 50% ethylene carbonate and 50% diethyl carbonate. A charge–discharge cycling test was performed by a Neware (Shenzhen, China) test system in the voltage range from 0.01 V to 3 V (vs. Li/Li^+^). Electrochemical impedance spectroscopy (EIS) was carried out in an electrochemical workstation (Chenhua, Shanghai, China) in the frequency range from 10 mHz to 100 kHz at a voltage amplitude of 5 mV. A cyclic voltammetry (CV) test was performed at a scanning rate of 0.1 mV s^−1^ in the voltage range from 3.0 V to 1.0 mV at room temperature.

## 3. Results and Discussion

Figure 1a shows the XRD patterns of C@Sn/NSGr-5, C@Sn/NSGr-7.5 and C@Sn/NSGr-10. The peaks near 26.3° and 44.4° correspond to the characteristic peaks of graphene at (002) and (101). The diffraction peaks at about 32.0°, 43.8° and 55.3° were assigned to the (101), (220) and (301) diffractions of Sn, corresponding to the crystal structure of Sn (JCPDS 04-0673). The partial diffraction peaks of Sn were not obvious, which might be due to the diffraction peaks of carbon and graphene being too prominent or overlapping. The results of Raman spectroscopy in Figure 1b display a D peak and a G peak at around 1345 cm^−1^ and 1580 cm^−1^, respectively. The D peak reflects the disordered structure caused by carbon defects, and the G peak indicates the degree of graphitization [32,33]. The results showed that the *I_D_*/*I_G_* values of C@Sn/NSGr-5, C@Sn/NSGr-7.5 and C@Sn/NSGr-10 were 0.28, 0.21 and 0.16, respectively, which indicated that the C@Sn/NSGr samples had good lattice structures. Figure 1c displays the nitrogen adsorption-desorption curves of the C@Sn/NSGr-5, C@Sn/NSGr-7.5 and C@Sn/NSGr-10 samples. They exhibit the typical characteristics of the type IV curve, indicating the existence of mesopores in the obtained materials [34]. Particularly, C@Sn/NSGr-7.5 demonstrated the maximum specific surface area of 100.8 m^2^ g^−1^, while the specific surface areas of C@Sn/NSGr-5 and C@Sn/NSGr-10 were 50.6 m^2^ g^−1^ and 96.7 m^2^ g^−1^, respectively. Meanwhile, the pore size distribution calculated by the BJH method showed that the pore size was mostly about 4.5 nm, as shown in Figure 1d [35], further confirming the samples possessed an abundant mesoporous structure. This structure has a large specific surface area and pore volume, a unique morphology, as well as excellent thermal and chemical properties [36]. Such considerable specific surface area and abundant mesopores contribute to creating a large contact area and promote the diffusion of lithium ions.

Figure 2 shows the morphological and structural features of graphene, C@Sn/NSGr-5, C@Sn/NSGr-7.5 and C@Sn/NSGr-10. It can be clearly seen that the obtained graphene presented a distinct thin layered structure, and the layers were cross-linked. Graphene displayed a wrinkled structure on the surface, which could be beneficial to increase the surface area and reaction sites of the materials, as shown in Figure 2a. Compared to materials without carbon coating (shown in Appendix A), Figure 2b–d show that all the obtained composites still maintained the layered structure of graphene, and there were uniformly distributed particles on the surface of the composites. In addition, after coating by glucose, the direct contact of the Sn particles was limited to some extent, which could effectively hinder Sn agglomeration. When the concentration of Sn^2+^ was low, the amount of Sn^2+^ diffusing on the surface of graphene was also low, resulting in only a small part of Sn^2+^ being reduced to metal Sn during the reduction process, so less metal Sn became attached to the surface of graphene. As the concentration of Sn^2+^ increased, more metal Sn was reduced on the graphene surface, as shown in Figure 1c,d (the Sn nanoparticles are marked with yellow circles). Therefore, this showed that Sn can be successfully plated on graphene by electroless Sn plating, and the agglomeration of Sn can be further limited after coating by glucose, finally reaching the effect of “double confinement” of metal Sn.

To further explore the microstructure of C@Sn/NSGr-7.5, TEM and HRTEM observations were implemented. It can be seen in Figure 3a,b that the Sn nanoparticles were evenly dispersed on graphene, without obvious agglomeration. The average diameter of the Sn nanoparticles was approximately 5 nm, which was beneficial to slowing the volume effect of Sn. Furthermore, Figure 3c shows that there were few layers of graphene in the composite and the metal Sn nanoparticles were coated with a carbon layer. In addition, Figure 3d shows that Sn lattice fringes were observed, with a lattice spacing of about 0.206 nm, corresponding to the (220) crystal plane of Sn. Figure 3f–j displays that the elements C, N, O, S and Sn were uniformly distributed in C@Sn/NSGr-7.5, and the content of each element was 88.13%, 0.42%, 5.37%, 0.20% and 6.30%, respectively, as shown in Appendix A. Therefore, these results proved that the electrochemical stripping method had successfully doped the sulfur and nitrogen elements into graphene and that the Sn nanoparticles were uniformly dispersed on graphene by electroless plating.

For investigating the composition and bonding state of the surface energy of the C@Sn/NSGr-7.5 sample, X-ray photoemission spectroscopy (XPS) was carried out. a clearly demonstrates the presence of C, Sn, O, N, and S elements in C@Sn/NSGr-7.5, further verifying that the electrochemical exfoliation process could effectively dope the nitrogen and sulfur elements. Figure 4b shows the high-resolution spectrum of C 1s. Peaks appeared at about 284.8 eV, 285.3 eV, 285.6 eV and 286.4 eV, corresponding to the C−C/C=C, C−S, C−N and C−O bonds, respectively [37]. In the high-resolution spectrum of Sn 3d (Figure 4c), The Sn 3d_5/2_ peak was split into three peaks with binding energies of 487.6 eV (Sn^4+^), 487 eV (Sn^2+^) and 485.2 eV (Sn), and the Sn 3d_3/2_ peak was split into three peaks with binding energies of 496.1 eV (Sn^4+^), 495.2 eV (Sn^2+^) and 493.8 eV (Sn). These results proved that electroless plating could reduce Sn^2+^ to metallic tin. There was a combination of Sn^4+^ and Sn^2+^ in the tin, which could be caused by the partial oxidation of metal Sn, consistent with the results of the O 1s spectrum [38,39]. Three peaks appear at about 531.8 eV, 534.0 eV and 531.0 eV in Figure 4d, which refer to the bonds of C=O, C−O and Sn−O−C, respectively. Figure 4e shows that three peaks were located at about 401.7 eV, 399.8 eV and 398.5eV, which corresponded to the bonds of graphitic N, pyridinic N and pyrrolic N [40], respectively. In the high resolution of S 2p, Figure 4f demonstrates that there were three peaks at around 163.7 eV, 165.2 eV and 168.8 eV, which were assigned to the chemical bond energies of C−S−C, C−S−C and C−SO_x_ [41,42], respectively. Noteworthily, nitrogen and sulfur co-doping contributed to increase the lithium storage capacity and promote charge transport and ion diffusion.

The CV curve of C@Sn/NSGr-7.5 in Figure 5a displays a peak at around 0.5 V during the first lithium insertion process which disappeared in the subsequent two cycles [23]. There was a sharp peak appearing near 0.01 V, which was attributed to the formation of the Li_x_Sn alloy and the insertion of lithium into the carbon layers to form Li_x_C [31,43]. In the following anodic process, the peaks appearing at about 0.25 V and 1.10 V indicated the processes of Li^+^ extraction from the carbon layers and Li_x_Sn conversion to Sn, respectively. Distinctly, similar and overlapping CV curves appeared in the subsequent two cycles, indicating a good electrochemical reversibility and stability of C@Sn/NSGr-7.5 during cathodic and anodic processes. Figure 5b–d presents the charge–discharge curves of C@Sn/NSGr-5, C@Sn/NSGr-7.5 and C@Sn/NSGr-10 at 0.1 A g^−1^. All the electrodes demonstrated a sloping voltage platform at around 1.00–0.25 V and 0.25–0.01 V during the initial lithiation process, which was related to the formation of Li_x_Sn and the insertion of Li^+^ into the carbon layers and graphene, respectively. During the initial delithiation process, there was a sloping voltage platform at 0.50–1.10 V in all composite materials, in connection with the extraction of Li^+^ from Li_x_Sn. In the subsequent cycles, all the electrodes demonstrated the disappearance of the discharge voltage platform of the first cycle (around 0.50–0.80 V), which could be due to the formation of a stable SEI film. These above results are consistent with the results of CV in Figure 5a. In addition, it can be seen that the initial discharge specific capacity of C@Sn/NSGr-5, C@Sn/NSGr-7.5 and C@Sn/NSGr-10 were 1436 mAh g^−1^, 1346 mAh g^−1^ and 1294 mAh g^−1^, corresponding to the charge specific capacity of 658 mAh g^−1^, 692 mAh g^−1^ and 694 mAh g^−1^, respectively. This huge loss of specific capacity might be caused by the formation of SEI or the decomposition of the electrolyte. Although the initial coulombic efficiency of all electrodes was low, it could reach about 95% during the subsequent cycles, indicating that the electrodes tended to be stable and possessed a very high cyclic reversibility. To verify the capacity of the C@Sn/SNGr-7.5 composite, the theoretical capacity of C@Sn/SNGr (Sn/C = 6.3:88.13 wt%) was calculated to be about 390 mAh g^−1^ based on the theoretical capacity of tin (994 mAh g^−1^) and the theoretical capacity of graphene and carbon (≈372 mAh g^−1^) [44]. The prepared composite materials exceed the theoretical capacity of 390.5 mAh g^−1^. This could be due to the fact that the composite material had a considerable mesoporous structure and presented sulfur and nitrogen co-doping, which are beneficial to improve the lithium storage capacity of the composite materials [45,46].

Figure 6a exhibits the cycle performances of C@Sn/NSGr-5, C@Sn/NSGr-7.5 and C@Sn/NSGr-10 at the current density of 0.1 A g^−1^. The specific capacity of the C@Sn/NSGr-5 electrode was low, which was due to the low content of Sn in the composite. As the concentration of stannous chloride increased, the specific capacity of the composite also increased. Distinctly, the C@Sn/NSGr-7.5 electrode delivered a higher reversible capacity than C@Sn/NSGr-5 and C@Sn/NSGr-10. The specific capacity of the composites coated by carbon was higher than that of the uncoated composites, as shown in Appendix A. Particularly, the C@Sn/NSGr-7.5 electrode had a good cycling performance, which could be due to the good synergistic effects of heteroatom-doped graphene, carbon coating and appropriate Sn concentration. The huge loss of specific capacity during the first charge–discharge process could be caused by the formation of the SEI or the decomposition of the electrolyte. With a continuous circulation, the electrolyte constantly wetted the composite material, promoting the lithium ions to be embedded in the deep interior of the composite materials, thus improving the lithium storage capacity of the composite materials. Meanwhile, the doping of sulfur and nitrogen induced defects in the graphene, increasing the insertion sites of lithium and providing a higher charge capacity during the cycles [45]. Figure 6b demonstrates the rate performance of C@Sn/NSGr-5, C@Sn/NSGr-7.5 and C@Sn/NSGr-10 at different current densities from 0.1 to 5 A g^−1^ and finally back to 0.1 A g^−1^. It can be seen that all the electrodes were relatively stable at each rate. Particularly, the C@Sn/NSGr-7.5 electrode displayed the best rate performance at each current density, and the reversible charge capacity was 663 mAh g^−1^, 516 mAh g^−1^, 389 mAh g^−1^, 290 mAh g^−1^, 209 mAh g^−1^ and 141 mAh g^−1^, respectively. When the current density was restored to 0.1 A g^−1^ again, C@Sn/NSGr-7.5 still displayed a high reversible charge capacity of 665 mAh g^−1^, proving that it possessed a high rate performance and good cyclic reversibility. Figure 6c demonstrates the electrochemical impedance spectra of C@Sn/NSGr-5, C@Sn/NSGr-7.5 and C@Sn/NSGr-10. All the electrodes showed similar Nyquist plots with a semicircle in the high-frequency range and a sloping straight line in the low-frequency range, attributed to the charge transfer resistance (R_ct_) and the Warburg resistance (Z_w_), respectively. It can be obviously seen that the charge transfer resistance value of the C@Sn/NSGr-7.5 electrode (355 Ω) was the smallest, compared to those of the C@Sn/NSGr-5 (474 Ω) and C@Sn/NSGr-10 electrodes (368 Ω), suggesting that the C@Sn/NSGr-7.5 electrode had a higher electron conductivity. All the C@Sn/NSGr electrodes exhibited good electrical conductivity, which could be due to the Sn nanoparticles shortening the transport path of the electrons, helping to improve their transport capacity. Sulfur and nitrogen doping in graphene could increase the speed of electron/ion diffusion and transport, which could the improve the reaction kinetics of composite materials [30,45]. Meanwhile, the good synergistic effect of carbon coating, sulfur and nitrogen co-doping of graphene and Sn nanoparticles could also improve the conductivity of the composite materials. Furthermore, according to the linear relationship between Z′ and ω^−1/2^, the C@Sn/NSGr-7.5 electrode displayed a slope of 231.49, which was lower than that of the C@Sn/NSGr-5 (1341.54) and C@Sn/NSGr-10 electrodes (527.01), suggesting that the C@Sn/NSGr-7.5 electrode possessed the faster interface kinetics of Li^+^. According to the formulas Z′ = R_e_ + R_ct_ + σω^−1/2^ and D = (R^2^T^2^)/(2A^2^n^4^F^4^C^2^σ^2^), the ion diffusion coefficient of C@Sn/NSGr-7.5 (9.519 × 10^−13^ cm^2^ s^−1^) was higher than those of C@Sn/NSGr-5 (2.834 × 10^−14^ cm^2^ s^−1^) and C@Sn/NSGr-10 (1.837 × 10^−13^ cm^2^ s^−1^), which further proved that the C@Sn/NSGr-7.5 electrode had a better electrochemical performance. Moreover, Appendix A shows the Nyquist plots and the corresponding value of σ before and after five cycles, which indicate that a considerable electronic conductivity of C@Sn/NSGr-7.5 could be obtained during the cycling process. This could be due to the increased number of lithium storage sites after cycling [47]. Nitrogen and sulfur doping in graphene improves the electronic conductivity, increases the rate of electron/ion diffusion and transport, and improves the reaction kinetics [30,45].

Figure 7a shows the CV curves at different scan rates, from 0.1 to 1.0 mV s^−1^ to explore the reaction kinetics of the C@Sn/NSGr-7.5 electrode. It can be seen that there is a pair of redox peaks at about 0.26 and 0.01 V in the CV curves. The capacitive effect could be calculated according to following equation [48]:i = av^b^(1)
where a and b refer to adjustable parameters, and i and v represent the current density and the scanning speed, respectively. Generally, when the value of b tends to 1, the capacitive behavior is dominant. Figure 7b shows that the values of b were 0.51 and 0.72 at about 0.26 V (peak 1) and 0.01 V (peak 2), respectively, which indicated the C@Sn/NSGr-7.5 electrode allowed diffusion-controlled and capacitance-controlled electrochemical reaction processes [49,50]. In addition, an equation was used to study the capacitive effects (k_1_v) and the diffusion-controlled insertion process (k_2_v^1/2^) of C@Sn/NSGr-7.5:i(V) = k_1v_ + k_2v_^1/2^(2)
where v refers to the scan rate. Usually, the value of k_1_ and k_2_ could confirm the ratio of the current, which is associated with the surface capacitance and sodium ion semi-infinite linear diffusion. Figure 7c shows that the C@Sn/NSGr-7.5 electrode provided a 71% (orange) capacitive contribution to the total capacity at a scan rate of 1 mV s^−1^, which is in agreement with the results in Figure 7b. Figure 7d shows that the capacitive charge storage contribution increased with the increase of the sweep rate from 0.1 to 1 mV s^−1^, reaching 45%, 48%, 61%, 64%, 66% and 71% of the total capacity. These results convincingly confirmed that the C@Sn/NSGr-7.5 electrode possesses diffusion-controlled and capacitance-controlled lithium storage ability and may improve the electrochemical performance of batteries [51].

## 4. Conclusions

In summary, we presented a high-efficiency preparation of C@Sn/NSGr for LIBs through a facile electrochemical exfoliation, electroless plating and carbon coating associated strategy. The results showed that the Sn nanoparticles were evenly distributed on the surface of graphene, without obvious agglomeration, indicating graphene and carbon coating can effectively limited the agglomeration of Sn. Moreover, the optimized composite material (C@Sn/NSGr-7.5) demonstrated a high specific capacity of about 667 mAh g^−1^ after 150 cycles at the current density of 0.1 A g^−1^, exhibiting good cycle performance. It displayed a good rate performance with a specific capacity of 663 mAh g^−1^, 516 mAh g^−1^, 389 mAh g^−1^, 290 mAh g^−1^, 209 mAh g^−1^ and 141 mAh g^−1^ at 0.1 A g^−1^, 0.2 A g^−1^, 0.5 A g^−1^, 1 A g^−1^, 2 A g^−1^ and 5 A g^−1^, respectively. Due to the synergistic effect of the high specific capacity of Sn and the excellent cycling stability of graphene and carbon, the C@Sn/NSGr composites exhibited an excellent electrochemical performance. Therefore, this study provides an efficient preparation approach to improve the electrochemical properties of tin-based anode materials for LIBs.

## Data Availability

Not applicable.

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
