# Peer review of "Rational Construction of C@Sn/NSGr Composites as Enhanced Performance Anodes for Lithium Ion Batteries"

_nanomaterials, 2023, doi:10.3390/nano13020271_

Round 1

Reviewer 1 Report

The manuscript by Yang et al. reports on the fabrication and electrochemical performance of composite material (C@Sn/NSGr) used as anode material in Li-ion batteries. This work, proposes a composite material, that is constructed of sulfur and nitrogen co-doped graphene decorated with metal tin nanoparticles covered with a carbon shell. The optimized composite material (C@Sn/NSGr-7.5) maintains a specific capacity of about 667 mAh g-1 after 150 cycles at the current density of 0.1 A g-1, and exhibits good cycle performance.

Tin shows a high specific capacity but exhibit also “volume effect” problems which cause reduction of the specific capacity, rate and cycle performance.  This work is interesting, but some issues have to clarified in order to be clearer. I suggest considering the following points:

11. The introduction is focused only in the graphite, tin and graphene and not in other types of anode materials.

22. In the discussion about the Figure 1a it is said that "The partial diffraction peaks of Sn are not obvious, which may be due to the low concentration of Sn or the overlap of the carbon and graphene". It is not clear to me the term “partial” and also the elemental mapping reveals 6.3% Sn. It is not so small this percentage.

33. Figure 1d: “The result exhibits the typical characteristics of the type IV curve in the obtained materials, indicating the existence of mesopores structure”. What are the features for a porous structure and how they look-like if they are not.

4 4. Figure 1d. “Meanwhile, Fig. 1d shows that pore diameter distribution of obtained materials focuses on about 4.5 nm, further confirming the samples possess an abundant mesoporous”. How do you calculate the mean porous size from this diagram?

55. How do you recognize experimentally the carbon shell of the Sn nanoparticles? It is not clear to me.

66. There is a discussion on the Figure 5a, what is the origin of the peaks. According to the text these are attributed to some alloys (LixSn and LixC). Is there any experimental evidence for the presence of these alloys after the cycles? How the composite look-like after 100 cycles?

Author Response

Thank you especially for your comments on our work. These comments have been valuable in revising and improving our paper and have been an important guide to our research. We have carefully revised the manuscript based on your comments, and the revised manuscript is marked in red.

Reviewer 2 Report

Authors synthesize carbon-coated Sn-nanoparticles sandwiched between N-and S-doped Graphene layers (C@Sn/NSGr) as an anode material with a high specific capacity of about 667 mAh/g after 150 cycles at the current density of 0.1 A/g. It’s almost double when compared to the commercially used graphite. Sn-based nanocomposites are now well known and are expected to have higher capacities but are prone to aggregation and huge volume change. Here, the authors show a route to encapsulate Sn nanoparticles with carbon to prevent their agglomeration. The results are interesting, but I do have some comments that authors must address before I recommend it for publication.

Comment:

1)     One of the highlights of the paper is the N and S doping of graphene that is possibly playing a role in enhancing the specific capacity of the material as well as preventing agglomeration of carbon-coated Sn nanoparticles.  Here authors must explain: What is the role of N and S doping and why it is important in their material? What if un-doped graphene is used to synthesize the anode material, a comparison with the current material (C@Sn/NSGr-7.5) would be great.   As can be seen from the XPS results (Fig 4 of the manuscript) the presence of Sulphur is almost negligible compared to Nitrogen. Is Sulphur helpful in achieving the desired specific capacity?

2)     If possible, I would suggest authors perform an MD simulation to understand the interaction between C-coated Sn nanoparticles. Further DFT-based binding energy calculation could highlight the strength of Li interaction with C@Sn and with N and S dopants.

Minor Comments:

1)     In line 22: Please mention what 7.5 stands for in  C@Sn/NSGr-7.5.

Please carefully read the entire manuscript for typos. Some of them are as follows:

2)     Line 47: “about” is misspelled.

3)     Line 60: “machining performance”. 

4)     Line 151: “in in” printed twice.

5)     The resolution of the images is poor, specifically the XPS graphs (Figure 4). 

Author Response

(The authors gave the same response as above.)

Reviewer 3 Report

The article “Rational construction of C@Sn/SNGr composite as enhanced performance anodes for lithium-ion batteries” reports on a new composite of tin metal nanoparticles wrapped with graphene and a carbon shell. The novelty resides in the electroless synthesis of the tin metal nanoparticles, as opposed to previous literature.

However, the paper fails to compare their results with the relevant literature. A large number of reports on composites of tin metal nanoparticles and graphene or carbon shell has been published in the past few years, I suggest only a few that I found very close to the present study:

-          Chaoye Zhu et al J Mater Sci Tech 2021, 87, 18 (10.1016/j.jmst.2020.12.075)

-          Xiaobing Deng et al, Chem Phys Lett 2022, 806, 140062 (10.1016/j.cplett.2022.140062)

-          Zhiqiang Zhu et al, Nano Lett 2014, 14, 153 (10.1021/nl403631h)

-          Jian Wang et al, ACS Appl Mater Interfaces 2019, 11, 30500 (10.1021/acsami.9b10613)

-          Olga Riedel et al, ACS Appl Nano Mater 2019, 2, 3577 (10.1021/acsanm.9b00544)

-          Xucun Ye et al, Nano Lett 2019, 19, 1860 (10.1021/acs.nanolett.8b04944)

-          Shukai Ding et al, J Coll Interf Sci 2021, 589, 308 (10.1016/j.jcis.2020.12.086)

The authors must cite and detail a series of these papers or others of the same kind. It is even more important that the authors discuss the relevance of their own work in sight of the literature they cite. This is not the case in the present version of the article.

The experimental section is incomplete. Required data, such as the amount of graphite in the exfoliation step or the amount of solvent required in the preparation of the composite, are missing.

The interpretation of the data must be improved:

-          The discussion of SEM images in Figure 2 claim the presence of tin nanoparticles. I could not find the tin nanoparticles in Figures 2a, 2b and 2c. The authors can use close-up views and arrows.

-          The EDS analysis is used to show that S- and N-doping was efficient. However, as no control analysis is provided, the low amount of S and N present in the final product could stem from the graphite initially used for exfoliation.

-          The XPS data interpretation is incorrect and shows that the author does not understand the XPS technique properly. In particular, the two peaks in the Sn3d spectrum are due to the different spin states 3d3/2 and 3d5/2, but correspond to the same chemical species. The position of the 3d5/2 at 486.3eV would rather correspond to SnO or SnO2 than to Sn(0). The discussion of the S2p spectrum has to be corrected too.

-          Knowing the mass composition of the composites, a theoretical capacity can be calculated. The obtained capacity at the first cycle must be compared to this theoretical value. A quantitative comparison with the composites without carbon shell would also be of interest, taking into account the content in tin for each anode.

-          The discussion of the Voltage-Capacity plots on lines 250-260 is unclear. I don’t see the sloping platform and the flat platform in Figure 5bcd as indicated on line 252. Clarify.

-          On figure 6, the capacity of the composite anodes drop after the first cycle and then increases between cycle 40 and 140. Please comment.

-          The EIS analysis shows an interesting improvement in electronic and ionic conductivity for the composite with medium content in tin. However, it is unclear where this improvement arise. Please discuss.

Some english phrases are difficult to understand because of missing words. Note that the word "nanoalized" does not exist.

Author Response

(The authors gave the same response as above.)

Round 2

Reviewer 1 Report

To be published as it is now.

Reviewer 2 Report

Authors satisfactorily addressed all my comments. 

Thank You